

**Increase of bacterial community induced-tolerance to Cr in response to soil**
**properties and Cr level in the soil**
Claudia Campillo-Cora[a*], Daniel Arenas-Lago[a], Manuel Arias-Estévez[a], David
Fernández-Calviño[a]
a Departamento de Bioloxía Vexetal e Ciencia do Solo, Facultade de Ciencias, Universidade de Vigo, As
Lagoas s/n, 32004 Ourense, Spain. * corresponding author: ccampillo@uvigo.es



**ABSTRACT**

Chromium (Cr) pollution in soils is a global concern that should be assessed. Pollution Induced Community Tolerance (PICT) methodology is a highly sensitive tool that can directly indicate metal toxicity in the microbial community. Ten soils with a wide range of properties were spiked with 31.25, 62.5, 125, 250, 500, 1000 and 2000 mg Cr·kg$^{-1}$, in addition to the control. Bacterial growth (using [$^3$H]-leucine incorporation technique) was used to determine PICT, that is, whether bacterial communities developed tolerance in response to Cr additions to different soil types. Some bacterial communities did not grow normally at 1000 or 2000 mg Cr·kg$^{-1}$, probably due to high Cr toxicity, while others did. Regarding below 500 mg Cr·kg$^{-1}$, bacterial communities showed two responses depending on soil type: 7 of the 10 studied soils showed increased tolerance to Cr, while for the remaining 3 soils did not develop tolerance to Cr. Furthermore, the Cr level from which bacterial communities developed tolerance was dependent on the soil, i.e. Cr was more toxic in some of studied soils. The Cr effect on microbial communities was mainly determined by Dissolved Organic Carbon (DOC) and the fraction of Cr extracted with distilled water (H$_2$O-Cr) (R$^2$ = 95.6 %). Their effect on Cr in the soil might lead to an increase in toxicity (selection phase of PICT).

Keywords:

PICT (Pollution-Induced Community Tolerance), bacterial growth, Cr (chromium), dissolved organic carbon, metal bioavailability, risk assessment





## 1. Introduction

Chromium (Cr) is a highly toxic non-essential metal for microorganisms and plants, that may naturally occur at high concentrations from parent materials, e.g. serpentine rocks (Adriano, 2001; Cervantes et al., 2001). The world average content of Cr in soils is 60 mg·kg$^{-1}$, but in soils developed from mafic and volcanic rocks can reach up to 10000 mg·kg$^{-1}$ (Gonnelli and Renella, 2013). Cr contents up to 2879 and 3865 mg·kg$^{-1}$ were reported for serpentine soils in Galicia (NW Spain) and Albania, respectively, (Covelo et al., 2007; Shallari et al., 1998). Anthropogenic activities, e.g. metallurgical industry, also lead to Cr accumulation in soils (Kabata-Pendias, 2011). Up to 195, 88 and 6228 mg·kg$^{-1}$ Cr were found in urban, agricultural and industrial soils, respectively (Srinivasa Gowd et al., 2010; Wei and Yang, 2010). Speciation and adsorption on soil solid surfaces are the main processes controlling Cr toxicity in soils (Adriano, 2001; Shahid et al., 2017). Despite the various Cr oxidation states, Cr(III) and Cr(VI) are the most stable and common forms in soils. Cr(VI) is considered the most toxic form of Cr, while Cr (III) is less mobile, less toxic and presents mostly precipitated (Kabata-Pendias, 2011). The adsorption of Cr on soil solid surfaces depends on several factors, e.g. soil pH, clay content, organic matter or Fe hydroxides (Bolan and Thiagarajan, 2001; Bradl, 2004; Dias-Ferreira et al., 2015; Gonnelli and Renella, 2013; Kabata-Pendias, 2011).

In the assessment of metal pollution, the toxic metal effect on soil microorganisms should be considered, because of their key role in maintaining soil ecosystem functions (Nannipieri et al., 2003). Lower microbial diversity, enzymatic activity, C mineralization and microbial biomass were found in Cr-polluted soil in comparison to unpolluted soil (Dotaniya et al., 2017; He et al., 2016; Pradhan et al., 2019). The potential nitrification and microbial abundance were inhibited with the increase of Cr level in the soil (Zhang





et al., 2022). Bacterial diversity was negatively correlated with total and available Cr,
while microbial community structure was altered (Zhang et al., 2021). However,
sometimes differentiating if the microbial response is due to Cr toxicity or to soil
properties variation is a difficult task (Liu et al., 2019), in addition to the complex
biogeochemical behaviour of Cr in soils (Ao et al., 2022). Therefore, a microbial indicator
specifically related to Cr toxicity that reduces interference of other soil properties is
needed to assess the Cr toxicity, such as the Pollution Induced Community Tolerance
(PICT) methodology. PICT is a sensitive tool that can be used as a direct indicator of
metal toxicity in the microbial community (Blanck, 2002). PICT methodology is based
on the selective pressure that the metal exerts on a microbial community, which favoured
the proliferation of more tolerant species over the more sensitive ones. Thus, the microbial
community that was exposed to the pollutant should show higher tolerance than that of
the unexposed reference microbial community (Blanck, 2002; Tlili et al., 2016). PICT
methodology has been successfully applied to assess Cr pollution in soils or sediments
(Gong et al., 2002; Ipsilantis and Coyne, 2007; Ogilvie and Grant, 2008; Santás-Miguel
et al., 2021; Shi et al., 2002a, 2002b; Van Beelen et al., 2004). The microbial community
tolerance should be quantified in a short-term assay by a sensitive endpoint, such as
bacterial growth measured using [$^3$H]-leucine incorporation (Berg et al., 2012; Boivin et
al., 2006; Lekfeldt et al., 2014). Despite the high sensitivity and specificity, the PICT
methodology might present some difficulties, mainly due to the influence of soil
properties (Blanck, 2002; Lekfeldt et al., 2014). Shi et al. (2002b) found similar values of
PICT to Cr and Pb both at low and high Cr (263 g·kg$^{-1}$) and Pb (10000 mg·kg$^{-1}$) levels,
respectively, suggesting that different soils affected Cr and Pb bioavailability. Similarly,
Shi et al. (2002a) did not found bacterial community tolerance to Cr (or Pb), regardless
of exposure history to Cr (or Pb), suggesting that several factors (organic matter, pH,



redox potential) might influence metal availability. Boivin et al. (2006), Fernández-
Calviño et al. (2012) and Fernández-Calviño and Bååth (2016) also reported different
tolerance values to heavy metals in soils with similar values of metals but different soil
properties. Soil properties may affect PICT development due to effects on metals
speciation, adsorption and bioavailability (Bradl, 2004; Shahid et al., 2017).
We hypothesize that soil pollution with Cr induces the development of bacterial
community tolerance to Cr, but the magnitude of the increases depends on soil
physicochemical characteristics. Therefore, we aim to determine the induced bacterial
community tolerance to Cr in response to the addition of different Cr levels to 10 soils
with variable properties. We also aim to assess the importance of soil properties on the
increase of bacterial community tolerance to Cr.




**2. Materials and Methods**

*2.1 Soil samples*

Soil samples were the same used previously in Campillo-Cora et al. (2021a, 2020) to study Cr adsorption and fractionation in soils with different properties, mainly in terms of organic matter and pH. In brief, ten remote forest locations in Galicia (NW Spain) were selected to avoid heavy metal pollution. Locations were also selected to obtain soil samples with a range of different physicochemical properties (Macías-Vázquez and Calvo de Anta, 2009). Superficial soil samples (0-20 cm) were taken using an Edelman probe and, once in the laboratory, were air-dried, homogenized, sieved (2 mm mesh) and stored until analysis.

*2.2 Soil properties*

A detailed description of the chemical analysis is given in Campillo-Cora et al. (2020) and in Supplementary Information. The properties of the 10 soils can be found in Tables S1 and S2. In brief, soil samples presented a wide range of textures (19-71 % Sand, 13-67 % Silt, 14-32 % Clay). A wide range of soil $pH_W$ and $pH_K$ was found: 4.0-7.5 and 3.0-6.9, respectively. Similarly, OM oscillated between 10-29 %. A range from 2 to 29 $cmol_c \cdot kg^{-1}$ was obtained for eCEC. A large range was obtained for DOC: 0.14 to 0.70 $g \cdot kg^{-1}$. Chromium total content varied from 7 up to 394 $mg \cdot kg^{-1}$.

Adsorption constants determined from Freundlich and Langmuir models (batch experiments) are presented in Table S3, obtained from Campillo-Cora et al. (2020). The different Cr fractions from extractions using distilled water, $CaCl_2$ and DTPA are shown in Table S4, obtained from Campillo-Cora et al. (2021a)



*2.3 Experimental design and bacterial community tolerance to Cr determination*


Sieved soil samples were rewetted until reaching 60 – 80% of water holding capacity
(Meisner et al., 2013). To rewet, soil samples were spiked with seven Cr solutions (made
from $K_2Cr_2O_7$) and one of distilled water, to obtain the following final Cr levels in soils:
2000, 1000, 500, 250, 125, 62.5, 31.25 and 0 mg Cr·kg$^{-1}$ soil. Each Cr solution was added
separately and in triplicate, finally obtaining 240 microcosms (10 soils x 8 [Cr] x 3
replicates). Once soil samples were spiked with Cr, microcosms were incubated in the
dark at 22 ℃ for two months, to ensure the reactivation of bacterial communities (Meisner
et al., 2013).
After the incubation period, bacterial community tolerance to Cr was estimated
through the PICT methodology (Blanck, 2002). The homogenization-centrifugation
technique was performed to extract soil bacterial communities (Bååth, 1992). The
bacterial community tolerance to Cr was determined as previously for Cu (Fernández-
Calviño et al., 2011), with modifications based on suggestions by Lekfeldt et al. (2014).
For this purpose, each microcosm was distributed in three 50 mL centrifuge tubes and
MES buffer was added in a ratio 1:10 soil/buffer (20 Mm pH 6; 4-
Morpholineethanesulfonic acid, CAS no: 4432-31-9) (Lekfeldt et al., 2014). The
suspensions soil/MES were mixed using a multi-vortex at maximum intensity for 3 min.
This step was followed by low-speed centrifugation to remove most of the fungal biomass
(1000 x g, 10 min) (Bååth, 1994; Bååth et al., 2001; Rousk and Bååth, 2011). Soil
supernatants, i.e. bacterial suspensions, were filtered through glass wool and 1.5 mL
aliquots were transferred into 2 mL micro-centrifugation tubes. A volume of 0.15 mL of
different Cr concentrations (made from $K_2Cr_2O_7$) was added to micro-centrifugation
tubes, obtaining nine Cr concentrations (3.3 x 10$^{-4}$ to 10$^{-8}$ M) plus a blank (0.15 mL of
distilled water). Then, the $^3$H-leucine incorporation method was used to estimate bacterial



growth (Bååth et al., 2001). A volume of 0.2 µL [³H]Leu (37 MBq mL⁻¹ and 5.74 TBq
mmol⁻¹. Amersham) with non-labelled Leu (19.8 µL) was added to each tube, resulting
in 300 nM Leu in the bacterial suspensions. Bacterial suspensions were incubated for 8 h
at 22ºC. Bacterial growth was stopped with 75 µL of 100% trichloroacetic acid. The
washing procedure and subsequent radioactivity measurement were carried out according
to Bååth et al. (2001). Radioactivity was measured by liquid scintillation counting using
a Tri-Carb 2810 TR (PerkinElmer, USA)

*2.4 Data analysis*
*2.4.1 Estimation of bacterial community tolerance to Cr (log IC$_{50}$)*
A dose-response curve was obtained for each soil microcosm. To compare the dose-
response curves, i.e. inhibition curves, with each other, bacterial growth was expressed
as relative bacterial growth. For each inhibition curve, generally, the four lowest added
metal concentrations to bacterial suspensions not showed bacterial growth inhibition
(Figure 1). Thus, relative bacterial growth was calculated by dividing all bacterial growth
data by the average of results from the four lowest added metal concentrations (including
blank), obtaining comparable dose-response curves. From each dose-response curve, log
IC$_{50}$ was determined as a tolerance index, i.e. Cr concentration resulting in 50% inhibition
of bacterial community growth. Higher log IC$_{50}$ values mean higher bacterial community
tolerance to Cr, and lower log IC$_{50}$ values mean lower bacterial community tolerance to
Cr. Log IC$_{50}$ was calculated using the following logistic model (Fernández-Calviño et al.,

2011):

$Y = c/(1 + e^{b(X-a)})$                 (equation 1)




where $Y$ is the measured level of Leu incorporation, $c$ is the bacterial growth rate without
added Cr, $b$ is a slope parameter indicating the inhibition rate, $X$ is the logarithm of Cr
added, and $a$ is log $IC_{50}$.

To detect whether bacterial community tolerance increase from different studied

soils occurs, $\Delta$log $IC_{50}$ was determined as the difference between log $IC_{50}$ value from each
Cr level in soil (2000, 1000, 500, 250, 125, 62.5 or 31.25 mg Cr·kg$^{-1}$) and the control soil
(0 mg Cr·kg$^{-1}$). A difference of 0.3 was taken as a reference value to determine if bacterial
community tolerance increased since it represents twice the Cr concentration in terms of
added Cr to bacterial suspensions. If $\Delta$log $IC_{50}$ is higher than 0.3, we will consider an
increase in bacterial community tolerance to Cr (Fernández-Calviño and Bååth, 2016,

2013).


*2.4.2 Estimation of bacterial community tolerance increase to Cr (multiple linear*
*regression analyses)*
A multiple regression analysis, using the backward elimination method, was performed
to obtain an equation that allows estimating the increase in bacterial community tolerance
to Cr ($\Delta$log $IC_{50}$) from soil properties. As the inhibition curves for some soils did not fit
the logistic model (equation 1) for the highest Cr concentrations (1000 and 2000 mg·kg$^{-1}$),
$\Delta$log $IC_{50}$ from 500 mg·kg$^{-1}$ was used for estimations. Once the equation was
estimated, determining factors were verified: linearity, error independency, residues
homoscedasticity, residuals normality, autocorrelation, collinearity and presence of
outliers. All statistics were performed using IBM SPSS Statistics 25 software (IBM,
USA).





**3. Results and discussion**

*3.1 Bacterial community tolerance to Cr in Cr-polluted soils with different properties*

Figure 1 shows bacterial growth inhibition curves obtained for each microcosm. Generally, a sigmoid dose-response behaviour is observed in the inhibition curves, indicating that when the added Cr concentration to bacterial suspension was low, relative bacterial growth was close to 1, while decreased when the Cr concentration increased. Most of the bacterial growth data fitted the logistic model, obtaining $R^2 \geq 0.87$, (Table S5). However, some data from 1000 and 2000 mg Cr·kg$^{-1}$ did not fit the logistic model, i.e. bacterial populations were not able to normally grow probably due to high Cr toxicity. In the case of 2000 mg·kg$^{-1}$, bacterial populations only grew normally in 4 of the 10 studied soils, while at 1000 mg·kg$^{-1}$ they grew normally in 7 soils. These differences in bacterial growth for the same Cr levels may indicate the influence of soil properties on Cr availability, as was previously suggested by Van Beelen et al. (2004). They found tolerant communities to Cr(III) in polluted soils with high Cr levels (2894 mg·kg$^{-1}$) but also reported that microbial communities from soils polluted with 3935 mg Cr·kg$^{-1}$ did not show tolerance to Cr(III), suggesting the influence of soil properties on metal toxicity. Therefore, in order to determine which properties influence Cr toxicity, the data of 1000 and 2000 mg Cr·kg$^{-1}$ were not considered in the following analysis.

The log IC$_{50}$ values determined from inhibition curves using the logistic model (equation 1) are presented in Table 1. Bacterial community tolerance to Cr (log IC$_{50}$) greatly varied between soils, even in the reference soils with no added Cr, log IC$_{50}$ oscillated from -6.40 (S8) up to -3.88 (S6) (log units). The variation of bacterial community tolerance to Cr in the reference soils may be an indicator that the development of PICT is dependent on soil type. In addition, this bacterial community tolerance to Cr fluctuation in reference soils, together with the natural Cr content in soils (7 – 394 mg·kg$^-$



[1], Table S2), highlights the importance of selecting reference soils for PICT studies
(Campillo-Cora et al., 2022a; Campillo-Cora et al., 2021b). Likewise, when Cr was added
to soils, bacterial community tolerance to Cr varied greatly between soils with the same
Cr level. A range from -6.37 (S8) to -3.56 (S6) was determined for soils polluted with the
lowest Cr level in soil (31.25 mg Cr·kg$^{-1}$); from -6.27 (S8) to -3.79 (S7) for 62.5 mg
Cr·kg$^{-1}$; from -6.26 (S8) to -3.65 (S7) for 125 mg Cr·kg$^{-1}$; from -6.27 (S5) to -3.41 (S7)
for 250 mg Cr·kg$^{-1}$; and from -6.09 (S8) to -2.87 (S3) for 500 mg·kg$^{-1}$.

Overall, bacterial communities showed two different responses to Cr addition to

the soil (Figure 2): (1) bacterial communities of S1, S2, S3, S6, S7, S8 and S10 developed
tolerance in response to Cr additions; while (2) bacterial communities of S4, S5 and S9
did not develop tolerance following Cr addition to the soil. Based on the PICT hypothesis,
the bacterial community is first exposed to the metal (i.e. selection phase of PICT), and
if metal exerts toxicity, then the most sensitive organisms of the community will
disappear, while the tolerant ones will be favoured. Therefore, whether the microbial
community developed tolerance to Cr is a toxicity indicator. Later, the microbial
community tolerance is quantified through a second exposition to Cr (i.e. detection phase
of PICT) (Blanck, 2002; Tlili et al., 2016). Accordingly, Gong et al. (2002) and Ipsilantis
and Coyne (2007) reported an increase in bacterial community tolerance to Cr with
increasing Cr levels in soil and rhizosphere. Van Beelen et al. (2004) found that bacterial
community tolerance to Cr(VI) increased with increasing Cr in pore water. Ogilvie and
Grant (2008) determined a tendency to increase the bacterial community tolerance to Cr
when the Cr level increases in estuarine sediments. Our results showed that bacterial
community tolerance to Cr increased with increasing Cr levels in soils only in 7 of the 10
soils studied (Figure 2). However, our results showed that the Cr level in soil from which
bacterial communities developed tolerance to Cr varied depending on the soil ($\Delta$log IC$_{50}$





> 0.3). Bacterial communities from S7 and S10 showed an increased tolerance at 31.25
mg Cr·kg$^{-1}$, bacterial communities from S1 and S3 at 62.5 mg Cr·kg$^{-1}$, bacterial
communities from S2 and S8 at 250 mg Cr·kg$^{-1}$, and bacterial communities from S6 at
500 mg Cr·kg$^{-1}$. In other words, Cr was more toxic for bacterial communities depending
on soil type, following the sequence: S7, S10 > S1, S3 > S2, S8 > S6. In other soils, our
results show that microbial communities did not develop tolerance to Cr, even at high Cr
levels. For example, bacterial communities of S6 did not show tolerance to Cr even at
2000 mg·kg$^{-1}$ (Figure 2). Similarly, Shi et al. (2002b, 2002a) and Ipsilantis and Coyne
(2007) did not find tolerant microbial communities to Cr even at high Cr levels, from 447
up to 263000 mg Cr·kg$^{-1}$. Therefore, considering that Cr-pollution sometimes has no toxic
effect on microbial communities and that, in other cases, microbial communities are
affected by Cr from very low levels of Cr-pollution, including soil properties in the
assessment of Cr-pollution is highly recommended, as for other heavy metals (Campillo-
Cora et al., 2022b).

*3.2 Estimation of the increase in bacterial community tolerance to Cr as a function of soil*
*properties*
The bacterial community tolerance to metals may be influenced by several soil properties,
such as soil pH, clay content or organic matter content (Ogilvie and Grant, 2008; Shi et
al., 2002b). The effect of soil properties on bacterial community tolerance can occur in
soil (selection phase of PICT), or in the determination phase of PICT. The effect of the
soil properties in the selection phase occurs in the soil, i.e. the first time bacterial
communities are exposed to the metal. For example, Fernández-Calviño and Bååth (2016)
found that bacterial community tolerance to Cu was lower in vineyard soils with high pH
in comparison to more acid soils, as Cu toxicity was reduced. On the other hand, the effect



of soil properties may occur in the detection phase, i.e. confounding factors leading to
altered tolerance measures (Lekfeldt et al., 2014). For example, Fernández-Calviño et al.
(2011) reported that the measurement of PICT to Cu was altered because of the presence
of the finer soil fraction in the bacterial suspensions when Cu concentrations were added.
That is, the finer particles will bind part of the Cu added to bacterial suspensions, resulting
in lower available Cu, so higher Cu concentrations will be necessary to inhibit the
bacterial growth leading to apparent higher tolerance, i.e. overestimated bacterial
community tolerance to Cu.

The equation presented in Table 2 related the increase of bacterial community

tolerance to Cr ($\Delta$log IC$_{50}$) with soil properties, explaining 95.6 % of the data variance (p
< 0.001). Only $\Delta$log IC$_{50}$ for 500 mg Cr·kg$^{-1}$ were used. The increase of bacterial
community tolerance to Cr was estimated by using soil properties (p < 0.05): DOC and
extracted Cr using distilled water (H$_2$O-Cr). Figure 3 shows estimated $\Delta$log IC$_{50}$ versus
measured $\Delta$log IC$_{50}$, with a homogeneous distribution around the line 1:1 (R$^2$ = 0.95).

DOC showed a significant positive relationship with $\Delta$log IC$_{50}$ (p < 0.05; Table

2), i.e. when DOC increases, the bacterial community tolerance to Cr also increases. This
DOC effect might be a confounding factor in the detection phase of PICT, as was
previously reported for Cu (Campillo-Cora et al., 2021b; Lekfeldt et al., 2014). When
bacterial communities are extracted from soil, DOC is extracted too. Later, when Cu is
added to bacterial suspensions, Cu and DOC may bind together (Beesley et al., 2010),
reducing Cu bioavailability and altering bacterial community tolerance to Cr
(overestimation). Bérard et al. (2016) reported a similar effect for microbial community
tolerance to Pb measurements. However, in a previous study (Campillo-Cora et al.,
2022c), we found that when dissolved organic matter (DOM) increases on bacterial
suspensions, then bacterial community tolerance to Cr decreases, i.e. when DOM



increases in bacterial suspensions, Cr becomes more toxic to bacteria. Hence, the DOC
effect in Cr bioavailability in the detection phase should be discarded because of the
positive relationship with $\Delta\log IC_{50}$ (Table 2) and attributed to an effect in the selection
phase in soil. In the soil, however, when DOC is present, Cr(VI) may be reduced to
Cr(III), i.e. Cr toxicity decreases when DOC is present (Ao et al., 2022). If fact, the use
of organic amendments to reduce Cr toxicity in soils is very common (Abou Jaoude et
al., 2020; Mitchell et al., 2018; Yang et al., 2021). A hypothesis is that the presence of
DOC in soil enhanced the reduction of Cr(VI) to Cr(III) (Wittbrodt and Palmer, 1997),
but during this process free radicals may also be formed (Kotaś and Stasicka, 2000),
increasing general toxicity for bacterial communities (Campillo-Cora et al., 2022c). In
response to increased toxicity in soil, then bacterial communities showed tolerance to Cr.
Another hypothesis might be the ability of Cr(III) to coordinate various organic
compounds, leading to the inhibition of some metalloenzyme systems (Kotaś and
Stasicka, 2000), which might result in a more tolerant bacterial community.
The Cr fraction extracted with distilled water ($H_2O$-Cr) showed a positive
relationship with $\Delta\log IC_{50}$ ($p < 0.001$, Table 2). Usually, the soluble form of heavy metals
represents the soil solution metal content, which is the most mobile and bioavailable form
(Kabata-Pendias, 2011). In the vase of Cr, probably Cr(VI) (Ao et al., 2022). Thus, $H_2O$-
Cr exerts its effect in soil, during the selection phase. $H_2O$-Cr content in soil increases as
added Cr level in soils increases (Campillo-Cora et al., 2021a). Whether Cr exerts
toxicity, the most sensitive bacterial species were removed, while the tolerant ones
survived, resulting in a more tolerant community to Cr. Later, in the detection phase,
when bacterial growth is measured and Cr is added to bacterial suspensions, tolerant
bacteria allow greater Cr concentrations, leading to a higher tolerant community. Van
Beelen et al. (2004) found a significant increase in microbial community tolerance to





Cr(VI) with Cr(VI) pore-water concentration. Similarly, Fernández-Calviño and Bååth
(2016) reported a positive relationship between bacterial community tolerance increase
($\Delta\log IC_{50}$) to Cu versus water-soluble Cu concentrations logarithm ($R^2 = 0.79$). Kunito
et al. (1999) also determined a positive correlation between $IC_{50}$ values and soluble-
exchangeable Cu ($r = 0.76$), while total Cu did not show any significant relationship ($r =$
$0.013$, $p > 0.05$).

*3.3 Concluding remarks*
In the present study, we aimed to improve the PICT methodology for the assessment of
soil pollution, using bacterial growth as the endpoint. Dissolved organic carbon (DOC)
and the fraction of Cr extracted with distilled water ($H_2O$-Cr) were the main factors
controlling the Cr effect on microbial communities, determined by the increase of
bacterial community tolerance to Cr. The main selection pressure of Cr on the microbial
community presumably occurs in soil, i.e. the selection phase of PICT. In the case of
DOC, Cr became more toxic to bacterial communities as DOC increased in soil, leading
to an increase in bacterial community tolerance to Cr in response to toxicity. Secondly,
$H_2O$-Cr is related to the toxic and active form of Cr, probably Cr(VI), and the higher the
$H_2O$-Cr content in the soil, the higher the tolerance to Cr developed by bacterial
communities. The outcomes of this study may be helpful for normalising Cr toxicity
thresholds for soil with different properties. In addition, overestimations or
underestimations of Cr toxicity based on total or bioavailable Cr content may be avoided,
since soil properties should be considered during risk assessment.





**Acknowledgements**

This study has been funded by the Spanish Ministry of Economy and Competitiveness through the project CTM2015-73422-JIN (FEDER Funds). David Fernández-Calviño holds a Ramón y Cajal contract (RYC-2016-20411) financed by the Spanish Ministry of Economy, Industry and Competitiveness. Daniel Arenas-Lago thanks the Ministerio de Ciencia e Innovación of Spain and the University of Vigo for the postdoc grant Juan de la Cierva Incorporación 2019 (IJC2019-042235-I). Claudia Campillo-Cora holds a Predoctoral fellowship financed by Xunta de Galicia (ED481A-2020/084). Funding for open access charge: Universidade de Vigo/CISUG.



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



**Tables**

**Table 1**

Bacterial community tolerance (expressed as log $IC_{50}$) to different levels of Cr pollution

in the 10 studied soils (average ± SE)

| Cr (mg·kg⁻¹) | 2000 | 1000 | 500 | 250 | 125 | 62.5 | 31.25 | 0 |
|---|---|---|---|---|---|---|---|---|
| **Soil** | Log $IC_{50}$±error | Log $IC_{50}$±error | Log $IC_{50}$±error | Log $IC_{50}$±error | Log $IC_{50}$±error | Log $IC_{50}$±error | Log $IC_{50}$±error | Log $IC_{50}$±error |
| **S1** | -5.34±0.03 | -5.35±0.05 | -5.28±0.03 | -5.30±0.03 | -5.33±0.03 | -5.30±0.04 | -5.83±0.06 | -5.82±0.05 |
| **S2** | -4.04±0.24 | -4.55±0.42 | -4.61±0.21 | -4.68±0.41 | -4.78±0.43 | -4.70±0.21 | -4.81±0.19 | -5.02±0.13 |
| **S3** | * | * | -2.87±0.51 | -4.38±0.15 | -4.62±0.16 | -4.70±0.18 | -5.46±0.03 | -5.38±0.05 |
| **S4** | -5.85±0.08 | -5.76±0.05 | -5.80±0.07 | -5.69±0.05 | -5.66±0.04 | -5.68±0.04 | -5.90±0.08 | -5.66±0.07 |
| **S5** | * | -4.47±0.11 | -5.80±0.19 | -6.27±0.07 | -5.86±0.10 | -5.98±0.06 | -6.02±0.10 | -6.09±0.07 |
| **S6** | * | -3.47±0.06 | -3.38±0.08 | -4.48±0.13 | -4.18±0.16 | -3.97±0.12 | -3.56±0.23 | -3.88±0.11 |
| **S7** | * | -3.44±0.09 | -3.35±0.07 | -3.41±0.09 | -3.65±0.11 | -3.79±0.07 | -3.85±0.05 | -4.32±0.12 |
| **S8** | -3.63±0.13 | -6.03±0.06 | -6.09±0.09 | -5.90±0.09 | -6.26±0.04 | -6.27±0.03 | -6.37±0.07 | -6.40±0.15 |
| **S9** | * | * | -4.32±0.27 | -4.37±0.39 | -4.70±0.23 | -4.43±0.13 | -3.82±0.05 | -4.11±0.04 |
| **S10** | * | * | -4.75±0.13 | -4.64±0.09 | -4.48±0.09 | -4.69±0.09 | -4.76±0.04 | -5.16±0.07 |

*Unadjusted data



**Table 2**
The equation for estimating bacterial community tolerance increase to Cr ($\Delta$log IC$_{50(500-}$
$_{0)}$) was obtained by multiple regression analysis using all soil samples (n=10).

| Equation | F | *p*-value | Adjusted R$^2$ |
|---|---|---|---|
| $\Delta$log IC$_{50}$ = - (0.435 $\pm$ 0.148) + (1.445 $\pm$ 0.320) **DOC** <br> *(p=0.026)*      *(p=0.004)* | 87.309 | <0.001 | 0.956 |
|      + (0.018 $\pm$ 0.001) **H$_2$O-Cr** <br> *(p<0.001)* | | | |

DOC is dissolved organic carbon (g·kg$^{-1}$); H$_2$O-Cr is Cr extracted using H$_2$O. Values associated with the
independent variables are shown together with the standard errors ($\pm$). P-values associated with each
independent variable are shown below variables (in brackets)







**Figures**

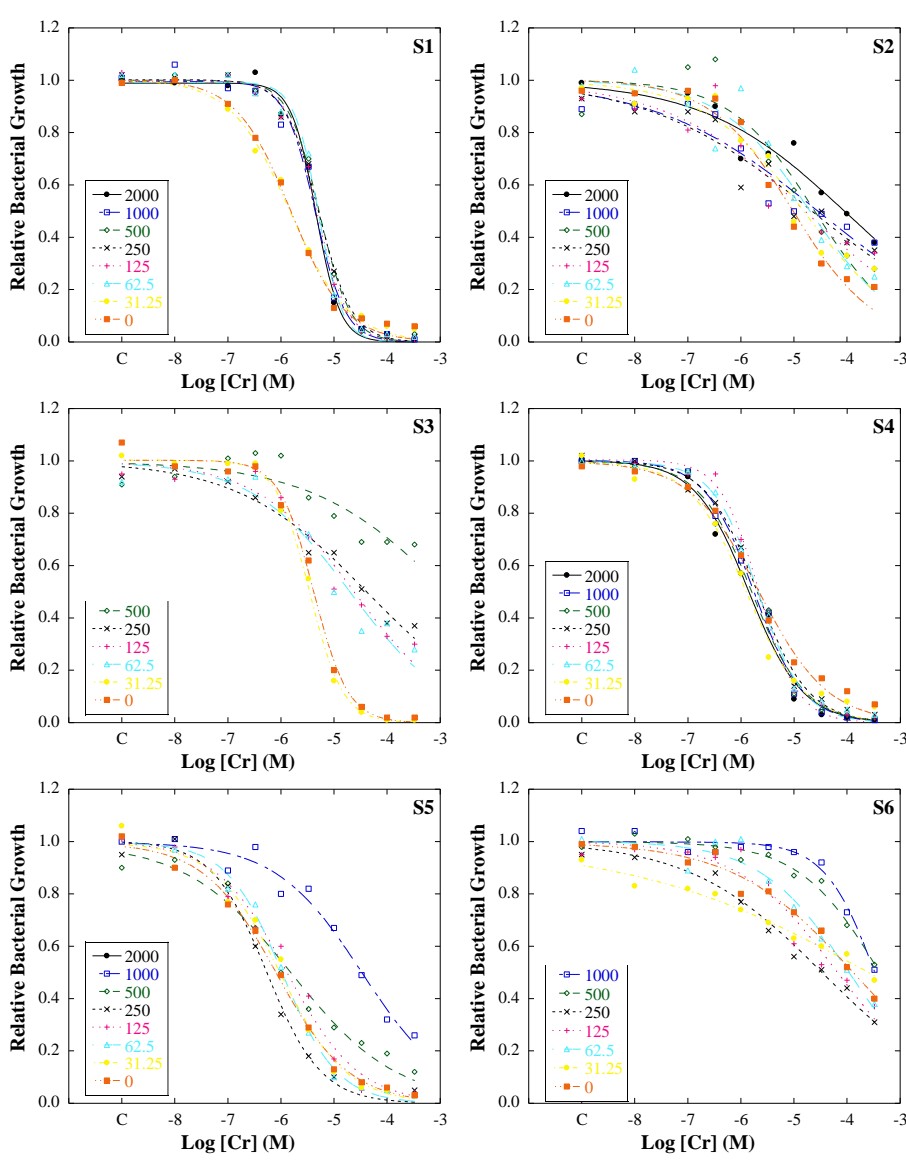

**Figure 1.** Bacterial growth inhibition curves for bacterial suspensions extracted from 10 soils artificially polluted with a range of Cr concentrations: 2000, 1000, 500, 125, 62.5, 31.25 and 0 mg·kg$^{-1}$. Dots indicate real data measured, while the lines represent the fit of the data to the logistic model used.



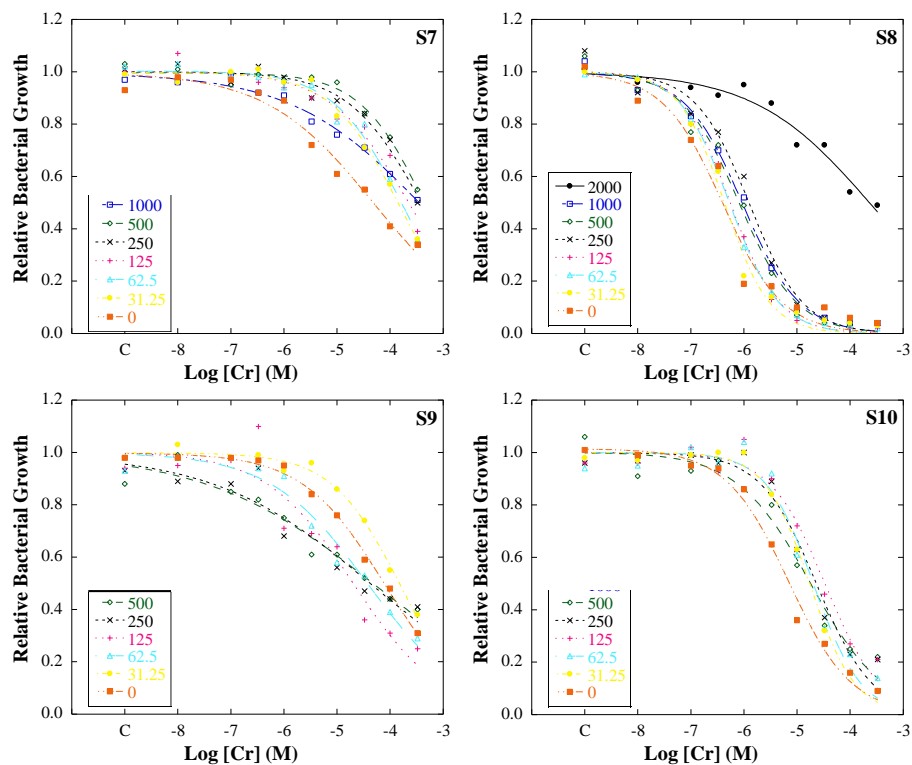


**Figure 1** (continued)



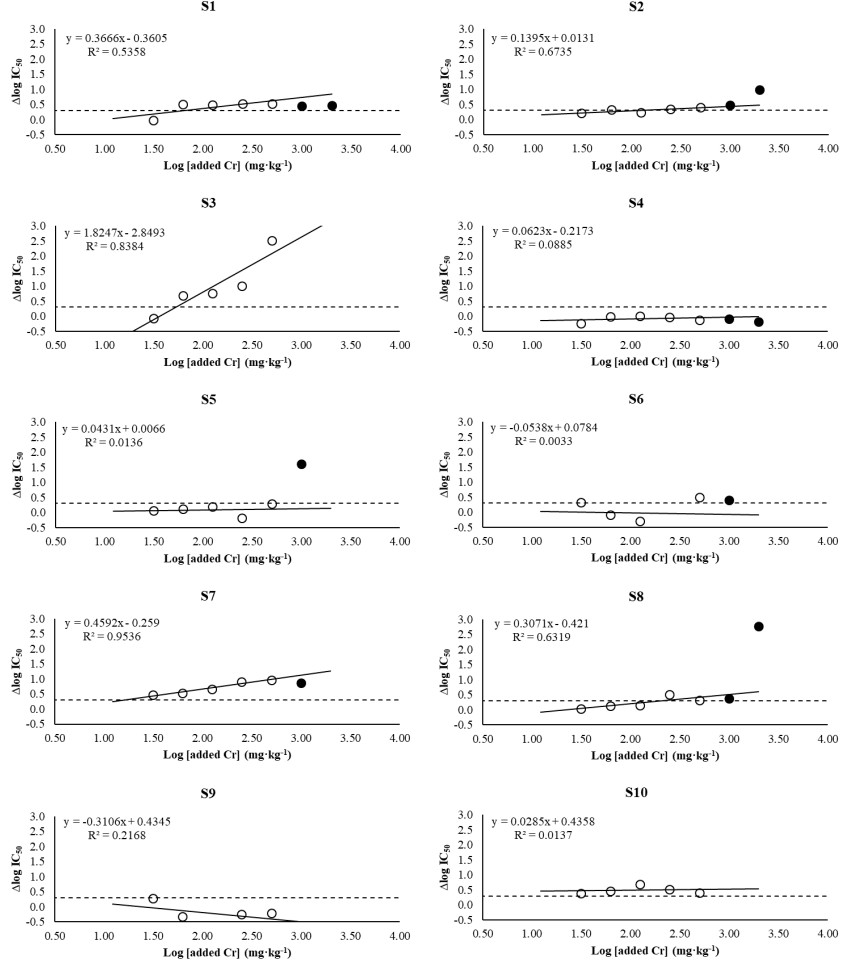


**Figure 2** Bacterial community tolerance variation (expressed as $\Delta$log IC$_{50}$ concerning

unpolluted soil) to a range of added Cr to soil (in logarithm scale). White dots represent

data from $\Delta$log IC$_{50(31.25-0)}$, $\Delta$log IC$_{50(62.5-0)}$, $\Delta$log IC$_{50(125-0)}$, $\Delta$log IC$_{50(250-0)}$ and $\Delta$log

IC$_{50(500-0)}$. Black dots represent data from $\Delta$log IC$_{50(1000-0)}$ and $\Delta$log IC$_{50(2000-0)}$. Continuous

lines represent linear regression fit. The discontinuous line represents the value (0.3) from

which it is considered that the bacterial community has developed tolerance.

589



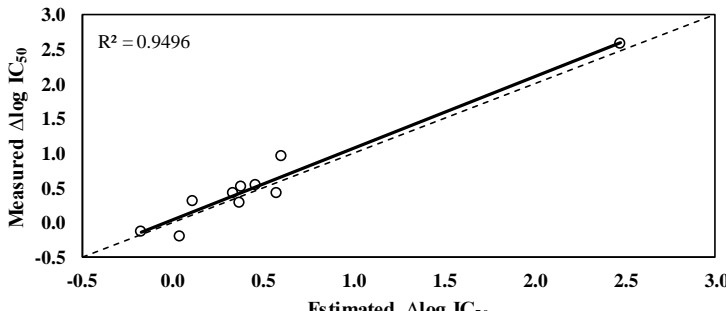

**Figure 3.** Relationship between measured and estimated $\Delta\log$ $IC_{50}$ using the equation

from Table 2.  The stippled line indicated a 1:1 relationship.