# Peer review of "Increase of bacterial community induced-tolerance to Cr in response to soil properties and Cr level in the soil"

_EGUsphere, 2023_

## Author Response (AR1)

You are kindly asked to individually respond to all referee comments (RCs) that have not yet been answered (marked in red). You can choose between posting a new author comment (AC) and co-listing an existing one in response to an RC. You are also invited to respond to other discussion contributions, if applicable.

**Referee #1**

In this study, the Pollution Induced Community Tolerance (PICT) method was used to investigate the effects of soil chromium pollution on microbial communities. The results showed that the tolerance level of bacterial communities to Cr was related to soil properties (DOC and H2O-Cr) and Cr pollution concentration. This study is helpful to standardize the threshold of chromium toxicity in soils of different properties. However, a number of issues should be carefully considered.

The detailed comments are presented as follows:

Why set the Cr levels as 2000, 1000, 500, 250, 125, 62.5, and 31.25 mg/kg? It should be described in the method.

> These concentrations were selected to encompass a broad range of contamination levels, which indeed follow an exponential scale. By working with such extensive ranges, it becomes more feasible to observe the development of bacterial community tolerance to Cr, taking into account the great variability in soil properties. This approach allowed for the comparison of results from soils with different properties. Furthermore, these were the concentrations previously utilized in the earlier studies of Campillo-Cora et al. (2020) (10.3390/agronomy10081113) and Campillo-Cora et al. (2021a) (10.3390/pr9061073).
>
> We added a little description of this explanation in L. 129 – 133.

Abstract should be carefully revised.

The Abstract has been substantially and carefully revised to enhance its overall quality.

Keywords: The abbreviation of PICT and Cr should be deleted.

Done

Line17: some bacterial communities, what is?

This sentence has been reformulated to enhance the quality of the abstract (L. 18 – 21)

Line20-21: what is 7 of the 10 studies soils, and the remaining 3 soils? Please specify it.

Done (L. 21 – 24)

Line22: dependent on the soil? What is? Please specify it.

Efforts are made to convey that the threshold of Cr concentration in the soils, at which bacterial communities develop tolerance to Cr, is not uniform but rather varies among soils. L. 24 – 26

Line24: the R2 for DOC?

The value of $R^2$ was obtained from a multiple linear regression analysis in which DOC and H2O-Cr were included as variables

Line25-26: Their effect on Cr….. Please modify this sentence.

Done (L. 28 – 30)

Line 73: You mentioned that PICT were widely to assess Cr in soil, what is the difference for your study?

It is not so much that it has been extensively applied, but rather that in the cited studies, PICT methodology has been successfully employed in soils and sediments. What distinguishes this study from those is the exploration of the soil properties (including Cr fractions and Cr sorption parameters) that exert the greatest influence on the development of bacterial community tolerance to Cr.

Line 109-120: Soil properties. It referred to original soil properties or?

Yes, it is referred to original soil properties.

For linear regression analysis of bacterial community tolerance to chromium, it is recommended to add some literature.

Done L. 191. Unfortunately, there are not many studies that have conducted a multiple linear regression between bacterial community tolerance to Cr and soil properties. Therefore, the references we have included correspond to some of our previous studies in which a similar data treatment was performed with Cr and other heavy metals (Cu, Ni, Pb and Zn).

DOC and H2O-Cr in soil properties were found to be related to the effects of Cr on microbial communities, so were other physicochemical factors taken into account?

Yes, all the measured physicochemical soil properties, Cr fractions in soils, and soil sorption parameters have been taken into account. However, it was the DOC and H2O-Cr that emerged as the most statistically significant variables and, therefore, were the ones selected.

Kindly supplement the amount of Cr(VI) and Cr(III) in the soil to support your conclusion.

Unfortunately, due to the incubation times required for this study, we do not have sufficient time available to carry out the measurements of Cr(III) and Cr(VI) to support our conclusions. One potential alternative could involve estimating the Cr speciation in the soil using Visual MINTEQ. However, it is important to note that the only physicochemical soil parameters that can be considered for such an estimation are the soil pH and the DOC content. This limited data would omit other properties that might intrinsically impact Cr speciation, considering that soil is a complex matrix. Therefore, estimation of Cr species by VisualMINTEQ software would not be feasible either. However, this suggestios will be duly considered for future research.

Line 312: Please explain what it means "In the vase of Cr, probably Cr (VI)"?

My apologies, there was a wording error that has now been rectified (L. 316)

Why use H2O-Cr, does not discuss CaCl2-Cr and DTPA-Cr, and other Cr fractions.

This was because the H20-Cr fraction exhibited a stronger statistical correlation with the development of bacterial community tolerance to Cr. Furthermore,

regarding the previous article where the fractions were extracted using CaCl2 and DTPA (Campillo-Cora et al. (2021a) (10.3390/pr9061073)), it was determined that the H2O-Cr fraction was the most abundant in comparison with CaCl2 and DTPA fraction in studied soils.

How does the abundance of bacterial communities change under different soil properties? Kindly provide additional data, such as 16 S rRNA sequencing, in this regard and discussion it.

It would be highly intriguing to investigate how the abundance of bacterial communities may vary in soils with different properties. However, that was nor the primary focus of this study, which centred on the development of bacterial community tolerance to Cr and its relationship with soil properties. Nonetheless, it could be compelling to initiate a new research avenue dedicated to exploring variations in bacterial community abundance in response to soil properties variation in future studies.

Fig 1 and Fig 2: Kindly add a note so that the reader knows exactly what the numbers S1, S2, etc. stand for.

Done L. 584 – 585 and 594 – 595.

Fig 1: Kingly adjust the legend for S6 and S7 because it covers the X-axis.

Done

Check the style of the reference.

Done, the style of the reference was corrected.

**Referee #2**

**MS No.: egusphere-2023-185**

General comments

The pollution-induced community tolerance (PICT) method is a highly sensitive tool that can directly reflect the metal toxicity of soil microbial communities, and has been widely

used in the study of soil heavy metal pollution (including Cr, Ni, Pb, Zn, and Cu). Although this study has some significance, its results do not bring enough new knowledge and lack of novelty. For example, how is the novelty of this study different from the following research?

[1] Campillo-Cora, Claudia, et al. Estimation of baseline levels of bacterial community tolerance to Cr, Ni, Pb, and Zn in unpolluted soils, a background for PICT (pollution-induced community tolerance) determination. Biology and Fertility of Soils, 2022, 58, 49-61.

[2] Campillo-Cora, Claudia, et al. Influence of soil properties on the development of bacterial community tolerance to Cu, Ni, Pb and Zn. Environmental Research, 2022, 214, 113920.

[3] Campillo-Cora, Claudia, et al. Bacterial community tolerance to Cu in soils with geochemical baseline concentrations (GBCs) of heavy metals: Importance for pollution induced community tolerance (PICT) determinations using the leucine incorporation method. Soil Biology and Biochemistry, 2021, 155, 108157.

> In previous studies, the PICT methodology was applied to unpolluted soils and soils contaminated with heavy metals; however, the objectives differed. In these studies, the PICT methodology was utilized to assess the development of bacterial community tolerance to heavy metals, serving as an indicator of the toxicity of that particular heavy metal.
>
> Nevertheless, one of the limitations of this methodology is that when applied to contaminated soils, there is absence of reference soils for comparison to determine whether bacterial communities have indeed developed tolerance to the heavy metal. Based on this consideration, the objectives of studies [1] and [3] were established, where bacterial community tolerance to Cr, Cu, Ni, Pb and Zn in uncontaminated soils was determined. Consequently, equations were obtained for each studied metal, enabling the estimation of the baseline level of bacterial community tolerance to these metals based on soils properties.

Article [2] shares similarities with the present study; however, a fundamental distinction exists. The heavy metals studies in [2] are typically found in soil as cations (Cu, Ni, Pb and Zn), whereas Cr in commonly present in anionic form, resulting in different behaviour in response to soil properties. Hence, Cr has been independently studied in this research.

Specific comments

Cr pollution in soils is a global concern that should be assessed. In this study, the authors selected 10 soil samples to solve this problem. As readers, we would like to know the following questions. (1) Why only 10 soil samples were selected? (2) What are the special properties of these 10 soil samples? (3) Where did the 10 soil samples come from? (4) Do the soil samples cover major soil types and ecosystems?

1) Only 10 soil samples were selected for this study due to the extensive volume of analysis required. In fact, a total of 2,400 measurements were conducted solely for bacterial growth samples (1 metal x 10 soils x 8 [Cr] x 10 [Cr in bacterial suspension] x 3 replicates), which constituted a substantial workload. Furthermore, these soils had been previously chosen for complementary research on metal adsorption in soils (Campillo-Cora et al., 2020) and Cr fractionation in soils (Campillo-Cora et al., 2021a).

2) /3) One of the key characteristics of these soils is their origin from remote forested areas in Galicia (NW Spain), chosen to minimize potential anthropogenic influences, as detailed in the supplementary material and previous articles by Campillo-Cora et al. (2020, 2021a). Additionally, the sampling locations were determined based on scientific literature, considering the parent materials of the soil (granite, schist, amphibolite, limestone), in order to maximize the variability of physicochemical soil properties, particularly soil pH and %C. Five of the selected soils exhibit a pH gradient ranging from 4.0 to 7.5, with a similar average %C of approximately 6.3%. Conversely, the remaining five soils selected showcase a %C gradient ranging from 3.4% to 14.3%C, along with a similar average pH of approximately 4.8. These physicochemical properties were chosen for soil selection, as they represent the primary soil characteristics that influence the speciation, and hence toxicity, of Cr.

4) This study serves as a preliminary investigation to ascertain whether there exists any relationship between the development of bacterial community tolerance to Cr and soil properties. To that end, 10 soils with diverse physicochemical properties, particularly with regard to pH and %C, were selected. It has been determined that there is indeed an influence between DOC, $H_2O$-Cr, and the development of bacterial community tolerance to Cr within the studied soils. We believe that it would be worthwhile to consider expanding this study in future research to encompass other soil types, ecosystems, or even soils developed from different parent materials.

In this study, the 10 soils with a wide range of properties were spiked with 31.25, 62.5, 125, 250, 500, 1000 and 2000 mg Cr kg-1, in addition to the control. Why did the authors set this series of Cr concentration gradients? Does this range of Cr concentrations apply to different soil types (e.g., acidic and calcareous soils) or to different ecosystems (e.g., agroecosystems and forest ecosystems)?

These Cr concentrations were selected to promote the development of bacterial community tolerance to Cr, as they represent a broad exponential range of Cr contamination levels. By employing this broad spectrum of Cr levels, it becomes more feasible to observe the development of bacterial community tolerance to Cr, considering the great variability of soil properties. Additionally, these concentrations align with those utilized in prior investigations by Campillo-Cora et al. (2020) (10.3390/agronomy10081113) and Campillo-Cora et al. (2021a) (10.3390/pr9061073).

Our study has confirmed that this range of concentrations is applicable to different soil types. For instance, the results have shown increased bacterial community tolerance to Cr both in acidic soils (S1, S2) and those developed on limestone (S6, S7). However, it is worth noting that this study exclusively focused on remote forest ecosystems to minimize human influence on the environment. Therefore, future research could explore different ecosystems, such as agroecosystems.

This study reveals the response of bacterial community induced-tolerance to soil properties and Cr levels based on the laboratory culture experiments, but can the results truly reflect the natural conditions? For example, does the bacterial culture model (soil samples were spiked with Cr, microcosms were incubated in the dark at 22 ºC for two months) in this study match the situation in situ?

Soil microcosms were incubated at 22ºC in darkness to ensure the reactivation of soil bacterial communities (following the recommendations of Meisner et al. (https://doi.org/10.1016/j.soilbio.2013.07.014)). This incubation period spanned two months to guarantee the induced development of bacterial community tolerance to Cr, considering that the soil was artificially contaminated in the laboratory. It is acknowledged that these conditions may not reflect field conditions; nevertheless, the results obtained serve as a preliminary foundation with controlled conditions for future comparisons to be made with in-situ samples in subsequent studies.